# Effect of Inhibin Immunization on Reproductive Hormones and Testicular Morphology of Dezhou Donkeys During the Non-Breeding Season

**DOI:** 10.3390/ani15060813

**Published:** 2025-03-13

**Authors:** Muhammad Faheem Akhtar, Muhammad Umar, Wenqiong Chai, Liangliang Li, Ejaz Ahmad, Changfa Wang

**Affiliations:** 1Liaocheng Research Institute of Donkey High-Efficiency Breeding and Ecological Feeding, Liaocheng University, Liaocheng 252059, China; 2Department of Animal Reproduction, Faculty of Veterinary and Animal Sciences, Lasbela University of Agriculture, Water and Marine Science, Uthal 90150, Pakistan; 3Department of Clinical Sciences, Faculty of Veterinary Science, Bahauddin Zakariya University, Multan 60800, Pakistan

**Keywords:** inhibin immunization, plasma hormone concentrations, spermatogenesis, non-breeding seasonality

## Abstract

In recent years, researchers have introduced several methods to improve the reproductive efficiency of donkeys. As a seasonal breeder, their reproduction is affected by seasonality. Among these advancements, inhibin immunization has emerged as a sublime biotechnological tool to improve the reproductive efficiency of animals. The donkey industry in China is booming, and with its development, there is a dire need to focus on all aspects of donkey farming including improving reproductive efficiency. In winter, semen collection is stopped due to lower semen quality and difficulty in semen collection. It not only lowers farm production but also lowers profits in terms of poor-quality semen. This study was an attempt to improve Dezhou donkey reproductive efficiency in winter months (November–February) by inhibin immunization. The results illustrated that INH immunization elevated plasma hormone concentrations of FSH, LH, AMH, and Activin A.

## 1. Introduction

China has a rich history of raising donkeys. In past years, the donkey population in China significantly declined. However, it is now increasing owing to the high nutritional value of meat, milk, and most of all, a very popular traditional Chinese medicine (E-jiao) that is prepared from donkey skin, which has various health-beneficial properties [1,2]. Among 24 donkey breeds in China, the Dezhou breed is quite popular in commercial farming due to its predominant black hair with a straight back and waist, arch of chest rib, and round, strong hooves [3,4]. With emerging technologies and advancements in the meat sector, the consumption of beef, mutton, and chicken meat has market value. However, the consumption of donkey meat and milk is indispensable. Dezhou donkey is further divided into Sanfen and Wutou breeds. They are covered with black hair; however, the Sanfen donkey has white hair on the eyes, around the nose, and under the belly [1]. Seasonal reproduction is usual in mammals [5]. The effect of seasonality on the reproductive efficiency of donkeys is highly controversial. It is influenced by various factors including breed, environment, nutrition, and health [6]. Long or short days highly influence the reproduction of seasonal breeders like donkeys. With approaching shorter day length having minimal sunlight, the semen quality of male animals also declines. Seasonal alterations affect hormonal regulation, gametogenesis, FSH, and testosterone secretions [7]. The testicular steroidogenesis, spermatogenesis, and endocrine functions of follicle-stimulating hormone (FSH) are linked with inhibin and activin [7]. Based on the available data, we can hypothesize that donkeys experience reproductive quiescence throughout the winter months when sunshine hours are reduced and that jacks’ reproductive efficacy is also downregulated during these times.

Inhibin (INH) is a glycoprotein hormone secreted by gonads (Sertoli cells in males and granulose cells in females). Inhibin is one of the major players on the hypothalamus-pituitary gonadal (HPG) axis [8]. Inhibin is a 31–34 kDa heterodimeric glycoprotein, which forms a disulfide-linked dimer that shares a common α-subunit and differs in β-subunit (βA-subunit and βB-subunit), βA in inhibin A (αβA) and βB in inhibin B (αβB). Inhibin is a member of the transforming growth factor β (TGF-β) superfamily and has been proposed as an autocrine/paracrine factor that modulates follicular growth, atresia, gonadotropin responsiveness, and steroidogenesis [8]. Inbibin is a negative feedback regulator of FSH, i.e., it down-regulates FSH secretion in males and females [9]. In most cases, active or passive INH immunization lowers the negative feedback effect of INH and results in the up-regulation of FSH [10]. In past years, immunization against inhibin proved to be a promising tool in improving the reproductive efficiency of animals. Inhibin immunization and progesterone (P4) treatment elevated ovarian follicle development, which will subsequently enhance the early embryo developments in Holstein cows [11]. Active immunization against INH improved the fresh and post-thaw semen quality of Beetal bucks [12]. Inhibin immunization at a dose of 0.5 mg greatly improved the diameter and size of pre-ovulatory follicles and ovarian follicles [13]. Immunization against INH improved fertility in cattle [14], up-regulated spermatogenesis and testicular development in rats [15], elevated testicular weights in Yangzhou goose ganders [8], and elevated FSH concentrations without affecting the concentration of LH and testosterone [16]. The potential for INH vaccination to raise FSH and testosterone levels is evident from these data. We hypothesized that INH immunization can elevate plasma hormone concentrations of FSH, LH, T, P4, AMH and Activin-A and spermatogenesis in months of low breeding in Dezhou donkeys. Therefore, the current study was designed to investigate the potential role of inhibin (INH) immunization during low-breeding seasonality (November–February) on plasma hormone concentrations of FSH, LH, T, P4 Progesterone, AMH, and Activin A including alterations in the testicular histoarchitecture 

## 2. Materials and Methods

### 2.1. Inhibin Immunogen Preparation

A porcine recombinant inhibin α-subunit protein was expressed in a prokaryotic expression system in *E*. *coli* strain BL21 (DE3) and was utilized as the inhibin antigen. The recombinant protein contained 175 amino acid residues, including a 41-residue leading sequence derived from the expression plasmid pRSETA (Invitrogen, Carlsbad, CA, USA). Also, it included 134-residue porcine inhibin α-subunit mature peptide. After purifying recombinant protein, it was homogenized with a mineral oil adjuvant, composed of a 1:2 (*v*/*v*) mixture of water and grade 10 white oil for injections (Hangzhou Refinery, Hangzhou, China), and the final concentration of immunogen was 1 mg/mL.

### 2.2. Experimental Design

This study was approved by the Research Committee of the Animal Policy and Welfare Committee of Liaocheng University (No. LC2019-1). The care and use of laboratory animals fully comply with local animal welfare laws, guidelines, and policies.

All animals were offered silage and had free access to drinking water. The current study was conducted from 27th November to 27th February (90 days) at Liaocheng Wanshixing Breeding Co., Ltd. (E 115° and N 36°), Liaocheng, Shandong Province, China. Adult Dezhou jacks, *n* = 30, with the same genetic origin and an average age of 2.5 ± 0.50 years, were randomly divided into three groups, i.e., A, B, and C. Each group had 10 animals each. All animals in experimental groups were intramuscularly (i.m) injected with different concentrations of inhibin. Animals in group A were immunized with 3 mg of inhibin immunogen; group B was immunized with 1.5 mg of inhibin; and group C was kept as the control group, receiving a bovine serum albumin (BSA) injection. Inhibin was injected on the 1st and 23rd days of the experiment in groups A and B. The dose of the first and booster shot of INH immunogen was 3 mg for group A and 1.5 mg for group B.

### 2.3. Body Weights

Throughout the experimental period (on day 1, 38th, 63rd, and 78th day of the experiment), body weights in all groups exhibited non-significant differences except on day 21, at which body weights of group A and C animals were significantly higher (*p* < 0.05) as compared to group B animals.

### 2.4. Measurement of Body Weight, Blood, and Testes Tissue Collection

Body weights of all animals in groups A, B, and C were measured on the 1st, 38th, 63rd, and 78th day of the experiment. Blood samples were collected on the 21st, 38th, 34th, and 40th day of the experiment via the jugular vein into heparinized tubes. Within three hours of sample collection, plasma was separated from the blood by centrifugation at 1000× *g* and kept at −20 °C until analysis. The testes tissues from groups A, B, and C were taken from the slaughterhouse at the end of the experiment. Testes were frozen in liquid nitrogen and kept at −80 °C as soon as they were collected.

### 2.5. Antibody Titer

Standard ELISA was used to analyze inhibin antibody titer in donkey plasma. The α -inhibin recombinant fusion protein was used to coat a 96-well microtiter plate (0.5 µg/well in 100 µL). Then, a 100 μL plasma sample (1:1200 dilutions with 5% skimmed milk) was added to each well and then incubated at room temperature to bind anti-inhibin antibodies with the coated inhibin fusion protein. The bound antibodies were further labeled by incubation with horseradish peroxidase (HRP) conjugated rabbit anti-bovine antibody (SantaCruzBiotechnology, SantaCruz, CA, USA). Finally, color development was preceded by chromogen tetramethyl Benzedrine (Sigma, West Hollywood, CA, USA) solution containing 0.03% H_2_O_2_, and terminated after appropriate with the addition of 2% H_2_SO_4_. Optical absorbance was taken at 450 nm on the EON Bioteke spectrophotometer to represent inhibin antibodies titer for both control and immunized donkeys. Figure 1 shows anti-inhibin antibody titres at 21st, 28th, 34th and 40th day of experiment.

### 2.6. Measurement of Plasma Hormone Concentrations

Plasma concentrations were determined by ELISA using quantitative kits (MEIMIAN from Jiangsu Meimian Industrial Co., Ltd., Nanjing, China). Assays were performed using protocols provided by the kit supplier. For FSH, assay sensitivity was 0.075 U/L. Both inter and intra-assay coefficients were below 10%. The detection range was 0.3 U/L–18 U/L. For plasma LH, assay sensitivity was 0.005 ng/mL. Inter and intra-assay coefficients were below 10%. The detection range was 0.002 ng/mL–0.05 ng/mL. For progesterone (P4), assay sensitivity was 5 pmol/L. Inter and intra-assay coefficients were below 10%. The detection range was 20 pmol/L–800 pmol/L. For testosterone, assay sensitivity was 0.02 ng/mL. Inter and intra-assay coefficients were below 10%. The detection range was 0.094 ng/mL–3.77 ng/mL. For AMH, assay sensitivity was 0.05 ng/mL. Inter and intra-assay coefficients were below 10%. The detection range was 0.2 ng/mL–8.5 ng/mL. For Activin (A), assay sensitivity was 0.4 ng/mL. Inter and intra-assay coefficients were below 10%. The detection range was 1.6 ng/mL–65 ng/mL.

### 2.7. Microscopy Performance

To observe the effect of INH immunization on testicular histoarchitecture, the testicular biopsy was performed on the 21st, 28th, 34th, and 40th day of an experiment by randomly selecting two animals from each group. The testicular biopsy was performed by the following procedure, as performed by Mohammadreza Baqerkhani et al. [17]. A slice of left testicular tissue (0.125 cm^3^) was taken and embedded in a 10% neutral buffered formalin solution for 24 h to observe histological alterations in seminiferous tubules. Histological analysis was performed with an automated tissue processor (LECIA RM 2235, Wetzlar, Germany). After fixation, tissues were dehydrated in alcohol of increasing concentrations, i.e., 70%, 80%, 90%, 100% and absolute alcohol. After dehydration, testis tissues were cleared in xylene embedded in paraffin wax. Testis tissues were cut perpendicular in 5 μm thickness to testicular long axis. Slides were then mounted on glass slides and were stained using hematoxylin and eosin (Nanjing Jiancheng Bioengineering Institute, Nanjing, China). Many histomorphometric metrics of the seminiferous epithelium have been measured, such as the diameter of seminiferous tubules, and; number of spermatogonia, spermatocytes, and elongated spermatids were calculated [18]. All ST were observed under a bright field light microscope (LEICA Dmi8, Wetzlar, Germany) with 40 X (25 μm) magnifications.

### 2.8. Criteria for Observing Apoptosis in Seminiferous Tubules

For observing apoptosis in seminiferous epithelium, the following criteria were followed and called Faheem’s score. We observed apoptosis in our previous work in Yangzhou ganders [18] and observed apoptosis in histological section slides of seminiferous epithelium after INH immunization as follows.

Seminiferous epithelium lumen seemed quite empty, showing degeneration of spermatogonia, spermatocytes, and apoptotic bodies.The Seminiferous tubule’s basal membrane was observed empty. Pyknotic germ and Sertoli cells were also observed.Sertoli cells vacuolated, and most tubules had empty lumen depicting impaired spermatogenesis.Seminiferous tubules had irregularly shaped and degenerated germ cells.

### 2.9. Statistical Analysis

The data were analyzed using SPSS (Version 20.0, Armonk, NY, USA) and Graph Pad Prism (Version 5.0). The Kolmogorov–Smirnov goodness-of-ft test was applied to determine normality. The data were transformed to logarithms if not normally distributed and then re-tested for normality before analysis. Then, a two-way ANOVA was applied to compare mean values. All the values were expressed as mean ± standard error of the mean (SEM). The differences across groups at various time points were analyzed Bonferroni post-test. The probability levels *p* < 0.05 or 0.001 were set to determine significant differences among groups.

## 3. Results

### 3.1. Anti-Inhibin Antibody Titer

On day 21, following the first inhibin immunization, OD values in group A initially did not exhibit any significant differences, as shown in Figure 1. Antibodies titers in groups B and A were substantially greater (*p* < 0.001) than those in the control group on the 34th day of the experiment. Titer of group B was 0.49, but somewhat lower (i.e., 0.40 of OD 450 nm). On day 40 of the experiment, groups A and B showed a similar pattern (*p* < 0.001). The OD = 450 nm was considered the base noise level of the assay.

### 3.2. Body Weight of Animals

Body weight of groups A and C were significantly higher as compared to group B on day 21, while it showed nin-significant difference on day 38th, 63rd and 78th day of experiment as shown in Figure 2. 

### 3.3. Plasma Hormone Concentrations

#### 3.3.1. Follicle-Stimulating Hormone (FSH)

Throughout the experimental period, the plasma FSH concentrations did not change significantly as shown in Figure 3. After 1st Inhibin immunization on day 1, the FSH concentration in group A was at a base level of 22 U/L on the 21st day of the experiment and it was up-regulated to peak levels of 29 U/L on the 28th day of the experiment and remained higher as compared to group B (1.5 mg INH) and C (Control) on 21 and 28th day of experiment, i.e., 15 U/L and 24 U/L in group B and 17 U/L and 27 U/L, respectively. After booster Inhibin immunization, FSH in group B remained at almost the same level, i.e., 22 U/L on the 38th day of the experiment but slightly elevated to 28 U/L on the 40th day of the experiment. Plasma FSH levels remained slightly higher on 21 and 28th day of the experiment, i.e., 17 U/L and 27 U/L as compared to group B.

#### 3.3.2. Luteinizing Hormone (LH)

Plasma hormone concentrations of LH (ng/mL) remained higher on the 21st and 28th day of the experiment in group A as compared to B and C, i.e., 0.021 (ng/mL) and 0.03 (ng/mL), respectively. Throughout the experimental period, the pattern of LH (ng/mL) concentration remained the same as between groups B and C even after booster immunization with inhibin shot on day 21. There existed non-significant differences among groups.

#### 3.3.3. Progesterone (P4)

Plasma progesterone (Pmol/L) concentrations displayed similar ascending and descending patterns throughout the experimental period except day 34, where P4 (Pmol/L) concentration was slightly elevated as compared to group C.

#### 3.3.4. Testosterone (T)

Plasma hormone concentrations of testosterone (T) remained higher, i.e., 1.8 (ng/mL) and 2 (mg/mL) in group A on days 21 and 28th as compared to groups B and C. On the 34th day of the experiment, the plasma T (ng/mL) in group was 2.2 (ng/mL), while it was 2 (mg/mL) in groups A and C. On the 40th day, plasma T (ng/mL) was almost the same, i.e., (2 ng/mL) in all groups. Moreover, plasma T (ng/mL) declined to 2 (ng/mL) on the 40th day as compared to 2.2 (ng/mL) in group B.

#### 3.3.5. AntiMullerian Hormone

Plasma AMH concentrations in group A were 3 (ng/mL) on day 21 in group A, while it was 3.85 (ng/mL) in group B. After booster INH immunization, it elevated to 5 ng/mL in group A and was almost 5 ng/mL in group B. On day 34, the AMH concentration was 4.9 ng/mL in group A, while it was slightly lowered in group B, i.e., 4.8 ng/mL. On day 40th of the experiment, plasma AMH was elevated to 5.9 (ng/mL) and it remained lower in groups A and C, i.e., 4.7 (ng/mL) and 4.2 (ng/mL), respectively. Throughout the experimental period, the plasma AMH concentrations remained lower in group C (control) as compared to groups A and B.

#### 3.3.6. Activin A

Plasma Activin A concentrations showed similar ascending and descending patterns to AMH throughout the experimental period. After the 1st INH immunization, plasma Actvin A concentration was 6 (ng/mL) in group A and 7 (ng/mL) in group B and 8 (ng/mL) in group C. On day 28th, it elevated to 11 (ng/mL) and 9 (ng/mL) in groups A, B and C, respectively. Activin A concentration increased significantly after the 2nd INH immunization on the 28th day as compared to the 21st day of the experiment. However, on day 40, their plasma Activin-A concentrations were the same in groups A, B, and C, i.e., 9 (ng/mL).

### 3.4. Germ Cell Count and Variations in Seminiferous Epithelium

From Figure 4, we can see that, throughout the experimental period, there existed non-significant differences among germ cells (spermatogonia, spermatocytes, and a number of elongated spermatids), except on day 28th, on which the number of spermatogonia and a number of elongated spermatids were significantly higher in group A as compared to group C. Also, empty lumen and apoptosis were observed in INH immunized groups (A and B) while normal lumen was observed in control group C.

## 4. Discussion

In the present study, the effect of INH immunization on the plasma hormone concentrations of FSH, LH, P4, testosterone, AMH, Activin A, anti-inhibin antibody titer and alterations in testicular histoarchitecture (seminiferous tubule diameter, and the number of germ cells, spermatogonia, spermatocytes, and elongated spermatids) in adult Dezhou donkeys (jacks) was determined. Our findings illustrated that INH immunization elevated plasma hormone concentrations of FSH, LH, AMH, and Activin A and caused apoptosis in the seminiferous epithelium. Throughout the experimental period, there existed a non-significant difference in ST diameter, no. of spermatocytes, spermatogonia, and elongated spermatids (except on the 28th day of the experiment). We also observed that the inhibin protein did not significantly affect the body weight, indicating modulation of inhibin levels at least within the parameters measured in this study does not influence overall body weight.

Previous studies in various species suggest that inhibin’s role has been predominantly focused on its involvement in reproductive functions and gonadotropin regulation, rather than energy metabolism or growth [19]. Moreover, gonads produce proteins in TGFβ family members (Inhibin A and B) that suppress the secretion of FSH without affecting LH secretion [20]. Conceivably, inhibin is primarily known for its role in inhibiting the secretion of follicle-stimulating hormone (FSH) and regulating the function of the gonads, rather than acting as a direct modulator of body weight or adiposity. Furthermore, inhibin is a glycoprotein hormone that is secreted by gonads (testes in males and ovaries in females) which inhibits the secretion and synthesis of FSH from the anterior pituitary gland [19]. Our findings also reinforce the theory that INH is a key modulator on the hypothalamus-pituitary gonadal axis only, rather than affecting metabolic activities in animals.

In the present study, inhibin antibody titer tended to increase in groups A and B after initial and booster INH immunization, indicating that INH immunization had triggered innate responses in these experimental groups. The reason may be that inhibin antibody titer is associated with testosterone concentrations and follows similar ascending and descending patterns in all groups. In our other study, though it was in Yanghou geese, inhibin antibody titer was associated with plasma testosterone concentrations [8].

Semen quality in male animals can be improved by enhancing plasma hormone concentrations of FSH. So, we can either enhance its secretion from FSH or mitigate its negative effects on testes. In our study, we attempted to elevate plasma FSH concentrations. In several studies, inhibin immunization has been proven as a promising tool in enhancing FSH secretion by downregulating negative feedback of endogenous inhibin [8,10,21,22,23,24,25]. In our outcomes, the FSH upregulation on the 21st and 28th days of the experiment enunciates that exogenous inhibin immunization had suppressed the effect of endogenous inhibin that resulted in elevated FSH secretion in group A (only on 21st day) as compared to groups B and C. On the hypothalamus–pituitary–testiscualr axis, FSH and testosterone secretion are downregulated or upregulated by inhibin, which plays a vital role in the up or down-regulation of spermatogenesis [26]. Moreover, inhibin immunization reduces the effect of endogenous inhibin and upregulates pituitary activity and FSH [14]. Furthermore, inhibin immunization also elevated the secretion of activin A [10,21,22]. Activin is the antagonist of inhibin, and elevated levels of activin may be due to activin secretion from Sertoli cells [27,28,29]. Both FSH and activin work together to regulate the function of Sertoli cells in the testes to regulate spermatogenesis [29,30,31]. Moreover, slightly upregulated FSH and activin may have stimulated Sertoli cell development. Similarly, FSH activates the cAMP/PKA pathway, which promotes the gene expression related to spermatogenesis and Sertoli cell function [32,33]. Ultimately, activin activates the Smad pathway, enhancing proliferation and survival [34,35]. To our knowledge, the present study speculates that both pathways integrate PI3K/Akt and MAPK/ERK signaling for Sertoli cell growth and differentiation.

In our present findings, plasma LH concentrations were higher after the first INH immunization, and after the first booster INH injection on day 28th, it was still higher among all groups but showed non-significant differences among all groups. At the same time, testosterone concentration followed ascending and descending patterns of plasma LH concentrations. This was initially a surprising result in light of other findings in which the LH concentration elevated after INH immunization in goats [13], dairy cows [21], and sows [36]. However, elevated levels of plasma LH after INH immunization can be species-specific. However, our previous findings elaborate that the LH-β did not increase significantly after INH immunization in Yangzhou ganders [8]. However, on the 34th and 40th day of the experiment, the plasma LH seemed to be slightly lower and upregulated and exhibited non-significant differences. It is also possible that INH immunization did not affect plasma LH concentrations and our results, INH did not improve plasma LH concentrations, significantly. In addition, the LHβ remains unaffected after INH immunization in aging white leghorn roosters [37]. As such, inhibin immunization did not affect the plasma LH concentrations in Shiba male goats [38]. On the hypothalamus pituitary gonadal axis (HPG), LH directly acts on Leydig cells that control T [39,40,41]. Our findings also reinforce previous outcomes.

Antimullerian hormone (AMH), a crucial marker of Sertoli cells, is a member of the TGF-β superfamily. AMH plays an important role in Sertoli cell development [42]. AMH has a vital function in testes development. According to our current findings, there was no discernible difference between experimental groups A (3 mg INH) and B (1.5 mg) and group C in the impact of INH immunization on plasma AMH.AMH is produced in fetal and postnatal Sertoli cells and its production lowers with puberty in adult males [43,44]. The development of Sertoli cells depends on the Sertoli cell marker Sox9 and it highly regulates the production of [45]. In our previous findings [8], we noticed that INH immunization did not affect gene expression levels of Sox9 and AMH in adult Yangzhou ganders. Our current findings are per the previous one in that, in adult animals, INH immunization does not affect AMH. So, we can speculate that if Sox9 regulation remains lower after INH immunization AMH must also remain lower.

Inhibin primarily acts to inhibit the secretion of follicle-stimulating hormone (FSH) and is thought to modulate gonadal function. In males, P4 is produced primarily in the testes by Leydig cells and Sertoli cells [46,47], where it serves as a precursor to other steroid hormones, including testosterone. It is possible that its effects on P4 production in male donkeys were not as pronounced as expected. Our findings suggest that, despite INH immunization, the P4 production pathways may be resistant to changes in FSH levels, or that compensatory mechanisms within the hypothalamic–pituitary–testicular axis may have prevented significant alterations in P4 concentrations.

Figure 4 enunciates alterations in testicular histoarchitecture in control group C and INH immunized groups A and B. In INH immunized groups A and B, there is a clear sign of apoptosis and Sertoli cell vacuolation. According to recent data, the phenomenon of steroidogenesis and breeding seasonality enhances the chance of apoptosis in the normal testis of seasonal breeder animals [48,49,50]. In our previous work, in which we had immunized Yangzhou gander with INH protein, similar findings were observed, i.e., apoptosis after INH immunization [8,18]. This may be possible due to the immunized group experiencing oxidative stress because of exogenous INH vaccination surpassing indigenous INH, which leads to high FSH levels and causes Sertoli cell vacuolation. Hormonal imbalance may disrupt testicular histoartichecture, including Sertoli cell vacuolation. Animals in the control group showed that some part of the lumen was empty too but there seemed no Sertoli cell vacuolation. However, signs of apoptotic cells may not be merely by INH immunization. Because they are seasonal breeders, there can also be a change in the testicular architecture that results in Sertoli cell vacuolation and apoptosis. Further, other studies reported that the histology of the testis alters with the breeding stage and age of maturity [18].

## 5. Conclusions

Taken together, inhibin immunization slightly upregulates plasma hormone concentrations of FSH, LH, testosterone, and AMH in nonbreeding seasonality in donkeys. Testicular histoarchitecture shows signs of apoptosis and Sertoli cell vacuolation. We speculate that inhibin immunization can also lower the efficiency of spermatogenesis in donkeys. The concomitant effect of non-breeding season and inhibin immunization caused imbalanced plasma hormone concentrations that resulted in disrupted testicular histoarchitecture. In contrast, the slight upregulation of FSH, LH, AMH, and testosterone after inhibin immunization gives the notion that if we slightly enhance the INH antigen dose, it may improve the reproductive efficiency of donkeys in terms of plasma hormone concentrations and semen quality. Semen-quality biomarkers and molecular pathways need to be explored for further studies.

## Figures and Tables

**Figure 1 animals-15-00813-f001:**
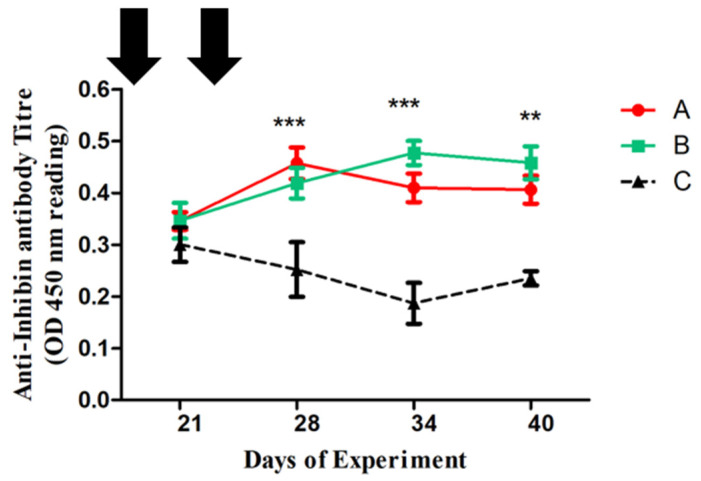
Anti-Inhibin antibody titers in Inhibin-immunized Group A 
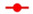
, Group B 
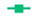
 and control Group C 
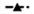
 of Dezhou donkeys at 21, 28, 34, and 40 days of the experiment. Vertical bars represent the standard error of the mean (SEM). The values with ** indicate the difference (*p* < 0.01) whereas the values with *** indicate the difference (*p* < 0.001) between groups A, B, and C. Arrows indicate primary and booster Inhibin (INH) immunization at 1st and 23rd day of the experiment. Each group (*n* = 10).

**Figure 2 animals-15-00813-f002:**
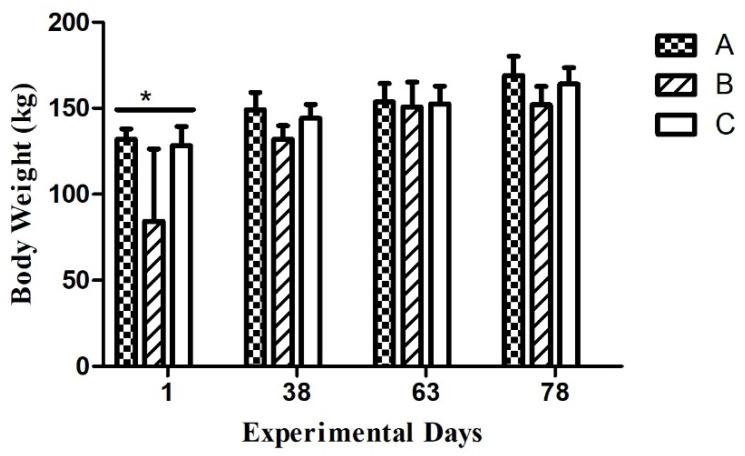
Dezhou donkey body weight on the 1st, 38th, 63rd, and 78th day of the experiment. Each bar represents the mean value of six determinations including the standard error. * indicate statistical significance based on *p* < 0.05 respectively.

**Figure 3 animals-15-00813-f003:**
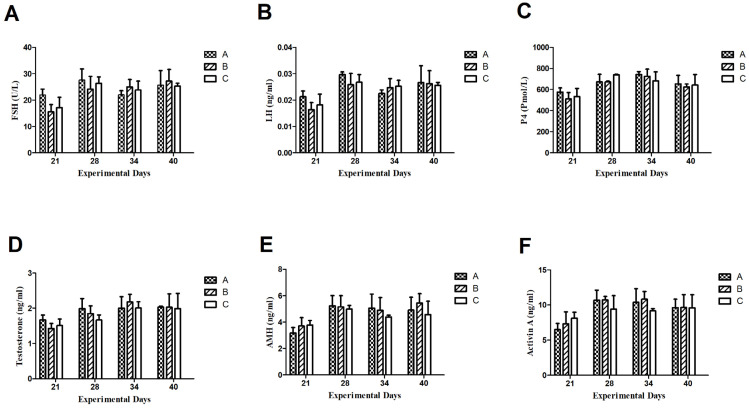
Plasma concentrations of Follicle-Stimulating Hormone (**A**), Luteining Hormone (**B**), progesterone (**C**), testosteterone (**D**), Anti-Mullerian hormone (**E**), and activin-A (**F**). Data are shown as mean values ± standard error of the mean. Each group (*n* = 10).

**Figure 4 animals-15-00813-f004:**
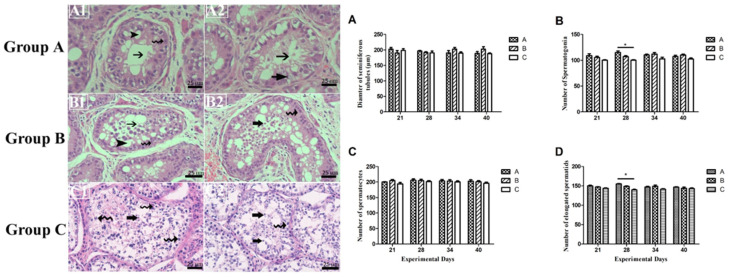
Histological sections from donkeys were collected on 21st, 28th, 34th, and 40th day of the experiment and stained with hematoxylin and eosin. Arrow with spiral tail: Spermatogonia, Arrowhead: Sertoli cell vacuolation, Arrow with tail: Empty lumen, Arrow with thick tail: elongated spermatids. Each group (*n* = 10) Two animals from each group were randomly selected for testicular biopsy for testes tissue collection. All images obeserved at 75 μm at Bar = 25×. Subfigures (**A**–**D**) depict morphometric measurements of histological sections and germ cell numbers. “*” Number of elongated spermatids were significantly higher as compared to group C. (**A1**) shows empty lumen and Sertoli cell vacuolation. (**B1**) showed sign of apoptosis in seminiferous epthelium and similar signs of apoptosis and sertoli cell vacuolation were observed in (**B2**); (**C1**) shows lumen filled with germ cells and spermatogenesis normal. (**A2**) shows some parts of lumen empty and spermatogenesis seemed to be abnormal.; (**C2**) shows normal lumen and spermatogenesis.

## Data Availability

The original contributions presented in this study are included in the article. Further inquiries can be directed to the corresponding author.

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
