# Peer review of "Effect of Inhibin Immunization on Reproductive Hormones and Testicular Morphology of Dezhou Donkeys During the Non-Breeding Season"

_animals, 2025, doi:10.3390/ani15060813_

Round 1

Reviewer 1 Report

Comments and Suggestions for Authors

The MS entitled “Effect of Inhibin Immunization...” by Akhtar et al attempted to investigate the effect of inhibin immunization in donkeys. Authors present the immunization responses especially in hormone levels and testicular histological morphology detected by ELISA and HE techniques.

General comment

The primary objective and the experimental design are basically feasible, however, the writing and interpreting of the results need improvement.   

The authors might have over interpreted the results: most of the results have no significant differences between the immunization and the control groups. In addition, two-way ANOVA is applied as in the materials and methods part, but no corresponding description in the results text or presentation in the figures.

Regarding the time points of samples collection, pls check for blood samples and body weight etc, the day 0/1 or before the immunization should be included.

The materials and methods are not clearly enough, e.g. How to evaluate the apoptosis in testicular histoarchitecture? Whats the sources of the ELISA kit?

Specific comment

The title is not appropriate, authors mainly detected hormone levels and testicular histological morphology which is related to but not the reproductive performance.

In the abstract, pay attention to the consistency: e.g. FSH, LH, AMH, and Activin A (Line 14, 27), FSH, LH, Testosterone, and Activin A (Line 23)

Line 24: spermatogonia significantly increased on 28th day

L37: itspredominant, need space

L48: on hormonal, redundant space

L69-74: authors can concise the writing from L69-74

L75-77: reorganize the hypothesis

L78: the objective needs to be clearer including the testicular histoarchitecture and hormone levels

L 99: figure 1, the 1st body weight (the first day body weight should be in the material parts rather than in the results; L118, BW also weighed on 21stt, 38th, 34th and 40th day of the experiment, but no results)?)? six determinations 111: Each group had 10 animals, only weight 6 animals?)? The statics is based on one -way ANOVA (Authors need to tell the groups comparison with *, ***)?

L122: kept at 20 ⁰C?

L126: The α-inhibin recombinant fusion protein, whats the species? How about the Inhibin protein homology with donkey?

L137: Whats the species of the FSH, LH, AMH and Activin-A Kit? How about the protein homology with donkey?

L153: 21, 28, 34th and 40th, Pay attention to the form

L162: In this part some indexes listed for determination, but no results for ST/field, luminal tubular diameter, epithelium height

L173: two-way ANOVA was applied, What the two factors are? Group and time? Therefore, authors need to show the significant marker in the result figure, respectively.

L181: no result for activin-A (F), also no ** and *** in this figure

L187: no result for day 21

L192: It then increased to 0.45 and 0.40 (P<0.001), mark the significance difference with symbols, respectively, in the figure. In addition, no need to repeat the data in the text. Same elsewhere

L200: Please write the significant results (with P value less than 0.05), do not repeat the data in the figure. Same elsewhere

L250: Whats differences between empty lumen and normal lumen? Whats the characteristics of apoptosis? How to detect the apoptosis? All these need to be described in the material and methods part.

L356: at 90th day of experiment? Please check the time. Please mark the basic characteristics for the tissue structure. Are there any changes for the interstitial e.g. Ledig cells? In addition, add the bar for each figure. Why the background for the groups is different? Seems not very consistent with the text in L245, and more detailed explanation is needed in the text part.

L256: Authors should tell the highlight of this study rather than “.... have thoroughly described in the first paragraph

L288: elevated FSH secretion in groups A only on 21Ist not 28th day

L303: plasma LH concentrations elevated after the first...  is significant? Please give a detail information in the figures. If correct, please explain it in the results and interpret (e.g. the current increase in ....) but not use repeated word in the discussion part. Same elsewhere

Comments on the Quality of English Language

The English could be improved to more clearly express the research.

Author Response

General comment

The primary objective and the experimental design are basically feasible, however, the writing and interpreting of the results need improvement.   

The authors might have over interpreted the results: most of the results have no significant differences between the immunization and the control groups. In addition, two-way ANOVA is applied as in the materials and methods part, but no corresponding description in the results text or presentation in the figures.

Regarding the time points of samples collection, pls check for blood samples and body weight etc, the day 0/1 or before the immunization should be included.

The materials and methods are not clearly enough, e.g. How to evaluate the apoptosis in testicular histoarchitecture? What’s the sources of the ELISA kit?

Response: Respected Reviewer, Thanks a lot for your valuable comments. Regarding apoptosis, from histological sections we had observed apoptotic cells visually, because this study included histological alterations after Inhibin immunization.

Specific comment

The title is not appropriate, authors mainly detected hormone levels and testicular histological morphology which is related to but not the reproductive performance.

Response: We have modified title “Effect of Inhibin Immunization on Reproductive Hormones and testicular morphology of Dezhou Donkeys During the Non-Breeding Season”.

In the abstract, pay attention to the consistency: e.g. FSH, LH, AMH, and Activin A (Line 14, 27), FSH, LH, Testosterone, and Activin A (Line 23)

Response: Corrected as suggested

Line 24: spermatogonia significantly increased on 28th day

Response: Number of spermtogonia in group A was significantly higher as compared to group C on 28th day of experiment.

L37: itspredominant, need space

Response: Corrected as suggested

L48: on hormonal, redundant space

Response:  Redundant space removed

L69-74: authors can concise the writing from L69-74

Response: Concised as suggested. “Inhibin immunization at the dose of 0.5 mg greatly improved the diameter and size of pre-ovulatory follicles and ovarian follicles [13], improved fertility in cattle [14],up-regulated  spermatogenesis and testicular development in rats [15]. elevated  testicular weights in Yangzhou goose ganders [8] and upregulated FSH concentrations without affecting the concentration of LH and testosterone.

L75-77: reorganize the hypothesis

Response: Reorganized as suggested “We hypothesized that INH immunization can elevated plasma hormones concentrations of FSH, LH, T, P4, AMH and Activin-A and spermatogenesis in months of low breeding in Dezhou donkeys.

L78: the objective needs to be clearer including the testicular histoarchitecture and hormone levels

Response: Corrected as suggested, “Therefore, the current study was designed to investigate the potential role of inhibin (INH) immunization during low-breeding seasonality (November-February) on plasma hormone concentrations of FSH, LH, T, P4 Progesterone, AMH, and Activin A including alterations in the testicular histoarchitecture.

L 99: figure 1, the 1st body weight (the first day body weight should be in the material parts rather than in the results; L118, BW also weighed on 21stt, 38th, 34th and 40th day of the experiment, but no results)?)? six determinations 111: Each group had 10 animals, only weight 6 animals?)? The statics is based on one -way ANOVA (Authors need to tell the groups comparison with *, ***)?

Response: Respected reviewer, body weights were only observed on Ist, 38th, 63rd and 78th day of experiment and I had written wrong in subheading 2.3, which I have corrected. In “Materials and methods” section, I have already written “2.3. Measurement of body weight, blood, and testes tissue collection” and explained about body weights. In Results section, under heading  “3.1. Body weights” already explain about body weights on respective days. All 10 animals in each group were weighed on days Ist, 38th, 63rd and 78th day of experiment. Yes we had performed one way ANOVA and where ever comparison result is *,*** , its mentioned in figure 1A.

L122: kept at 20 ⁰C?

Response: Sorry, corrected -20⁰C.

L126: The α-inhibin recombinant fusion protein, what’s the species? How about the Inhibin protein homology with donkey?

Response: Inhibin protein was prepared with E. coli BL21 (which is commonly used for recombinant protein expression) can be used in donkeys as well, given that inhibin is a conserved protein across many species. Inhibin's amino acid sequence is generally well-conserved among mammals, and this includes donkeys, cattle, goats, and pigs. Since donkeys are closely related to other equids (like horses), and inhibin is a critical hormone in regulating reproduction, so we expect a high degree of homology in the alpha and beta subunits across species.

L137: What’s the species of the FSH, LH, AMH and Activin-A Kit? How about the protein homology with donkey?

Response: Species of FSH, LH, AMH and Activin-A Kit was equine. The amino acid sequence homology between donkeys (Equus asinus) and the species for the ELISA kit was analyzed using BLASTp. The results indicate a high degree of conservation, with sequence identity above ≥85-90% in the functional domains. This suggests that the ELISA antibodies effectively recognize the target proteins in donkeys, supporting the validity of the assay results.

L153: 21, 28, 34th and 40th, Pay attention to the form

Response: Corrected as suggested

L162: In this part some indexes listed for determination, but no results for ST/field, luminal tubular diameter, epithelium height

Response: Corrected as suggested

L173: two-way ANOVA was applied, What the two factors are? Group and time? Therefore, authors need to show the significant marker in the result figure, respectively.

Response: Yes, two factors were groups and experimental days (time). I appreciate the reviewer’s request to highlight significant markers in the result figures. However, our statistical analysis did not yield significant differences

L181: no result for activin-A (F), also no ** and *** in this figure

Response: Sorry for missing activin-A results in figure 1 (F). Now I have added. I appreciate the reviewer’s request to highlight significant markers in the result figures. However, our statistical analysis did not yield significant differences

L187: no result for day 21

Response: Body weights were observed on day 1st, 38th, 63rd and 78th day of experiment. Not on 21Ist day of experiment. Body weights and blood samples collection days were different because its not appropriate to take blood samples of all animals and weighing them on weigh scale at same day, it causes stress to animals also.

L192: It then increased to 0.45 and 0.40 (P<0.001), mark the significance difference with symbols, respectively, in the figure. In addition, no need to repeat the data in the text. Same elsewhere

Response: I have already mark significance difference with symbols, respectively in figure 1B on day 28. I have deleted the sentence in text also as per suggestion.

L200: Please write the significant results (with P value less than 0.05), do not repeat the data in the figure. Same elsewhere

Response: I have clearly written, “Throughout the experimental period, the plasma FSH concentrations did not change significantly”. Difference was not significant…

L250: What’s differences between empty lumen and normal lumen? What’s the characteristics of apoptosis? How to detect the apoptosis? All these need to be described in the material and methods part.

Response: Normal lumen has spermatids, sperms and germ cells development under normal conditions while empty lumen is not normal under normal circumstances. I have written criteria of apoptosis in histological sections in Material and methods section

2.8. Criteria for observing Apoptosis in seminiferous tubules

L356: at 90th day of experiment? Please check the time. Please mark the basic characteristics for the tissue structure. Are there any changes for the interstitial e.g. Ledig cells? In addition, add the bar for each figure. Why the background for the groups is different? Seems not very consistent with the text in L245, and more detailed explanation is needed in the text part.

Response: Sorry for wrong imput. Histological sections from donkeys were collected on 21Ist, 28th, 34th and 40th day of experiment. There were no changes for the interstitial e.g. Ledig cells. Background of groups are bit different due to staining, nothing else. I have added clear bar for each picture also.

L256: Authors should tell the highlight of this study rather than “.... have thoroughly described” in the first paragraph

Response: We have highlighted first paragraph “ In the present study, effect of INH immunization on the plasma hormine concentrations of FSH, LH, P4, Testosterone, AMH, Activin A, anti-inhibin antibody titerand alterations in testicular histoarchitecture(seminiferous tubule diameter, and the number of germ cells ,spermatogonia, spermatocytes, and elongated spermatids) in adult Dezhou donkeys (jacks) was determined. Our findings illustrated that INH immunization elevated plasma hormone concentrations of FSH, LH, AMH, and Activin A and caused apoptosis in seminiferous epithelium. Throughout experimental perios, there existed non-significant difference in ST diameter, no. of spermatocytes, spermatogonia and elongated spermatids (except on 28th day of experiment) We also observed that the inhibin protein did not significantly affect the body weight, indicating modulation of inhibin levels at least within the parameters measured in this study does not influence overall body weight. 

L288: elevated FSH secretion in groups A only on 21Ist not 28th day

Response: Added “only on 21Ist day”

L303: plasma LH concentrations elevated after the first...  is significant? Please give a detail information in the figures. If correct, please explain it in the results and interpret (e.g. the current increase in ....) but not use repeated word in the discussion part. Same elsewhere

Response: In our present findings, plasma LH concentrations were higher after the first INH immunization, and after the first booster INH injection on day 28th, it was still higher among all groups, but showed non-significant difference among groups all groups

Reviewer 2 Report

Comments and Suggestions for Authors

The aim of the presented study was to check whether the injections of inhibin (twice) is able to change the plasma levels of several hormones conected with the pituitary-testis axis in the male donkeys during the non-breeding season. Authors also wanted to check the testicular histoarchitecture. The experiment was performed from November to February on 30 adult donkeys. Plasma levels of hormones-FSH, LH, progesterone, testosterone,AMH, activin A were not significantly changed during the experiment.

  1. Title should be changed because the term :"Reproductive Performance" is not clear. Maybe it will be better to change into: "Effect of Inhibin Immunization on Reproductive hormones and semen quality During the Non-Breeding Season in Dezhou Donkeys".
  2. Lack of obvious changes in the plasma hormones and no information about the semen quality did not confirmed the conclusion.
  3. Material and Methods should be clearly written-a/ how many days the experiment was lasting -90/78 ? when exactly the biopsy? or slaughter?was performed on only 2 animals? b/how many animals was in each group-10 or 6 (according  to the Fig. description); c/why was no blood drawing on day 1st (no results); d/ what was the dose of the second inhibin injection? e/ what was the plasma inhibin level after first and second injection? f/what was "mineral oil adjuvant"?Freund? 
  4. Description of the figures 1, 2 (no  part F), 3 and 4 must be checked and change to the appropriate.
  5. Why is lack of the semen parameters?
  6. Why was the blood sampling finished at 40 days?

The problem described in the manuscript is not novel because decreasing of the hormones level by antibody is known by decades. Also, the experimental model is characteristic and important to the local community -maybe it will serve as a example for creating another experiments.

In general the manuscript needs major revision.

Comments on the Quality of English Language

The English language should be improved by native speaker, manuscript has many errors which need to be corrected.

Author Response

Comments and Suggestions for Authors

The aim of the presented study was to check whether the injections of inhibin (twice) is able to change the plasma levels of several hormones conected with the pituitary-testis axis in the male donkeys during the non-breeding season. Authors also wanted to check the testicular histoarchitecture. The experiment was performed from November to February on 30 adult donkeys. Plasma levels of hormones-FSH, LH, progesterone, testosterone, AMH, activin A were not significantly changed during the experiment.

Response: Thanks a lot Respected Reviewer for your valuable comments and suggestions.

1. Title should be changed because the term:"Reproductive Performance" is not clear. Maybe it will be better to change into: "Effect of Inhibin Immunization on Reproductive hormones and semen quality During the Non-Breeding Season in Dezhou Donkeys".

Response: We have changed titled to “Effect of Inhibin Immunization on Reproductive hormones and testicular histoarchitecture during Non-Breeding Season in Dezhou Donkeys”

2. Lack of obvious changes in the plasma hormones and no information about the semen quality did not confirm the conclusion.

Response: Yes, plasma hormones concentrations had non-significant difference but stilled INH immunization elevated plasma hormone concentrations of FSH, LH, AMH, and Activin A. Moreover, semen collection was extremely difficult to perform due to harsh cold temperature and usually semen collection is stopped in these months also at farms also.

3. Material and Methods should be clearly written-a/ how many days the experiment was lasting -90/78 ? when exactly the biopsy? or slaughter?was performed on only 2 animals? b/how many animals was in each group-10 or 6 (according  to the Fig. description); c/why was no blood drawing on day 1st (no results); d/ what was the dose of the second inhibin injection? e/ what was the plasma inhibin level after first and second injection? f/what was "mineral oil adjuvant"?Freund? 

Response: In line 108, its clearly written that experiment lasted for 90 days.

(line 153-154) To observe the effect of INH immunization on testicular histoarchitecture, testicular biopsy was performed on 21, 28, 34th and 40th day of experiment by randonly selecting two aniamls from each group. Animals were not slaughtered for sampling.

b- There were 10 animals in each group.

c- Animals in groups A and B were injected with INH protein on day Ist while animals in group C were injected with BSA. It was not appropriate to have so much stress on all animals and to draw blood samples on day first and most of all blood samples collection in day Ist was not needed.

d- Dose of first and booster shot of INH immunogen was 3 mg for group A and 1.5 mg for group B. (I have also added at line 126)

e- Plasma inhibin levels were not analysed

f- mineral oil adjuvant (Solarbio Life Sciences) (Mineral oil is used in Freund's adjuvant)

Testicular biospy was only performed from 2 animals from each group as its not feasible to collect samples from all animals by testicular biopsy.

4. Description of the figures 1, 2 (no  part F), 3 and 4 must be checked and change to the appropriate.

Response: Figure F is added in Figure 2. Figures 3 and 4 are combined in Figure 3.

5. Why is lack of the semen parameters?

Response: Semen collection in non-seasonal breeding is not feasible. Semen collection is commercially stopped during non-breeding months. Later on we will design some plan to collect and analyze semen parameters in next studies.

6. Why was the blood sampling finished at 40 days?

Response: Inhibin immunogen antigen produce antibodies after 20 days of administration. We injected INH antigen on day first, and booster shot on 23rd day. So, blood samples were started from 21ist day and stopped at 40th day of experiment.

The problem described in the manuscript is not novel because decreasing of the hormones level by antibody is known by decades. Also, the experimental model is characteristic and important to the local community -maybe it will serve as a example for creating another experiments.

Response:  I agree, but here we checked effect of INH in non breeding season and no one has ever conducted on donkeys.

Reviewer 3 Report

Comments and Suggestions for Authors

The manuscript entitled 'Effect of Inhibin Immunization on Reproductive Performance During the Non-Breeding Season in Dezhou Donkeys' describes the effect of active immunization against inhibin on the levels of some hormones involved in the physiology of reproduction in male donkeys. The aim was to verify if there were margins of improvement in some reproductive aspects during periods that were far from reproductive activity.
The work is both interesting and effective in improving the reproductive management of donkeys, a species of veterinary interest that is often overlooked. There are some aspects, as reported below, that need to be clarified.

The experimental plan encompasses three study groups: two experimental groups with 1.5 and 3.0 mg of inhibin immunogen, and a control group that substituted the antigen with fetal bovine serum. I would like to know how you decided on the doses of inhibin immunogen. Furthermore, there were two administrations on the 1st and 23rd days. I would like to ask you why you chose this option, given that based on bibliographical findings, the administrations were generally more numerous. as a control you chose a group to which fetal bovine serum was administered: wouldn't it have been better to use, in addition, a control to which no administration was planned?

This study, and others that were conducted on other animals, have highlighted improvements in some reproductive aspects during the non-breeding season. Is this treatment capable of influencing the reproductive functions of the breeding season?
You have included figures 3 and 4 in the discussions. I would recommend moving them to the Results, in a separate paragraph where the histological aspects are described.
Minor remark: there are numerous typing errors. A very careful reading of the manuscript is necessary for corrections.

Comments on the Quality of English Language

Although the English language appears fluent, there are many typing errors (such as reversing letters and missing spaces).

Author Response

The manuscript entitled 'Effect of Inhibin Immunization on Reproductive Performance During the Non-Breeding Season in Dezhou Donkeys' describes the effect of active immunization against inhibin on the levels of some hormones involved in the physiology of reproduction in male donkeys. The aim was to verify if there were margins of improvement in some reproductive aspects during periods that were far from reproductive activity.
The work is both interesting and effective in improving the reproductive management of donkeys, a species of veterinary interest that is often overlooked. There are some aspects, as reported below, that need to be clarified.

The experimental plan encompasses three study groups: two experimental groups with 1.5 and 3.0 mg of inhibin immunogen, and a control group that substituted the antigen with fetal bovine serum. I would like to know how you decided on the doses of inhibin immunogen. Furthermore, there were two administrations on the 1st and 23rd days. I would like to ask you why you chose this option, given that based on bibliographical findings, the administrations were generally more numerous. as a control you chose a group to which fetal bovine serum was administered: wouldn't it have been better to use, in addition, a control to which no administration was planned?

Response: Thanks a lot, Respected Reviewer for your valuable comments. We had specified Inhibin immunogen dose on basis of litetrature and from our previous working experiments. From research papers

“Effects of immunization against inhibin α-subunit on ovarian structures, pregnancy rate, embryonic and fetal losses, and prolificacy rate in goats where estrus was induced during the non-breeding season”

“Long term effects of immunization against inhibin on fresh and post-thawed semen quality and sperm kinematics during low and peak breeding seasons in Beetal bucks”

“The role of active immunization against inhibin a-subunit on testicular development, testosterone concentration and relevant genes expressions in testis, hypothalamus and pituitary glands in Yangzhou goose ganders”

“Rectifying cow infertility under heat stress by immunization against inhibin and supplementation of progesterone”

                                   I had been done myself previously inhibin immunization work or was part of it. Seeing physiology of animal and body weight, high dose i.e. 3 mg and low dose 1.5 mg was decided.

This study, and others that were conducted on other animals, have highlighted improvements in some reproductive aspects during the non-breeding season. Is this treatment capable of influencing the reproductive functions of the breeding season?

Response: Respected reviewer, Inhibin immunogen at 3 mg and 1.5 mg doses, elevated plasma FSH, LH, AMH, and Activin A. Especially 3 mg Inhibin immunogen had better results. So on basis on of it, we can speculate that inhibin immunization will certainly improve reproductive functions of the breeding season.
You have included figures 3 and 4 in the discussions. I would recommend moving them to the Results, in a separate paragraph where the histological aspects are described.
Minor remark: there are numerous typing errors. A very careful reading of the manuscript is necessary for corrections.

Response: We have combined figures 3,4 into figure 3 and there is alreadt separate section in results section explaining histological aspects 3.3“Germ cells count and variations in Seminiferous epithelium”

Round 2

Reviewer 1 Report

Comments and Suggestions for Authors

Authors have modified the MS appropriately.

Regarding to the body weight, I prefer to being provided as the supplementary data, at least being placed after the methods 2.4. 

Comments on the Quality of English Language

Authors should check the MS to keep it consistent.

e.g.  63rd in Line 122 Vs68th in Line 127 

Line 162: 21Ist, should be 21st,

Line 217: 75 µm at 25X, pls use Bar=?

Author Response

Comments and Suggestions for Authors

Authors have modified the MS appropriately.

Respected Reviewer, thanks a lot for your valuable suggestions and positive response.

Regarding to the body weight, I prefer to being provided as the supplementary data, at least being placed after the methods 2.4. 

Response: corrected as suggested

Comments on the Quality of English Language

Authors should check the MS to keep it consistent.

e.g.  63rd in Line 122 Vs68th in Line 127 

Response: Sorry for inconvenience, corrected as suggested

Line 162: 21Ist, should be 21st,

Response: Corrected as suggested

Line 217: 75 µm at 25X, pls use Bar=?

Response: Corrected as suggested

Reviewer 2 Report

Comments and Suggestions for Authors

Dear Authors,

The manuscript needs careful checking by native speaker-it has many errors.

No further comments

Comments on the Quality of English Language

I have found many wrong written sentences and many letter errors

Author Response

Comments and Suggestions for Authors

Dear Authors,

The manuscript needs careful checking by native speaker-it has many errors.

No further comments

Response: Thanks a lot, Respected reviewer for your valuable comments

Comments on the Quality of English Language

I have found many wrong written sentences and many letter errors

Response: I have proof read whole manuscript and tried my best to correct and improve quality of english

Submission Date

26 January 2025

Date of this review

28 Feb 2025 14:52:57

Reviewer 3 Report

Comments and Suggestions for Authors

Dear Authors,
You have answered my comments satisfactorily.
Before I give my manuscript my approval for publication, there are a few more minor revisions that I would ask you to correct.
Figure 1: clarify whether p<0.0001 (in both A and B) corresponds to ** or ***?
In the title "of" is cited twice.
L. 174: delete exuberant punctuation.
L. 178: "follwing" should be "following".
L. 183: Speratogonia should be.
L. 216-217: Clarify, please.
L. 279: Perios?
L. 326: delete "groups", the first one.
These are some examples of typos, but there are probably more. So, I strongly recommend doing a serious and thorough check.

Author Response

Comments and Suggestions for Authors

Dear Authors,
You have answered my comments satisfactorily.
Before I give my manuscript my approval for publication, there are a few more minor revisions that I would ask you to correct.
Comment: Figure 1: clarify whether p<0.0001 (in both A and B) corresponds to ** or ***?

Response: P < 0.05 corresponds to * and P < 0.001 correspods to ***
Comment: In the title "of" is cited twice.

Response: Corrected as suggested
Comment: L. 174: delete exuberant punctuation.

Response: Correced as suggested
Comment: L. 178: "follwing" should be "following".

Response: Corrected as suggested
Comment: L. 183: Speratogonia should be.

Response: Corrected as suggested
Comment: L. 216-217: Clarify, please.

Response: These graphs of germ cells were drawn after counting all germ cell numbers on specific days of experiments and they enunciate non-significant difference among them
Comment: L. 279: Perios?

Response: Corrected as suggested
Comment: L. 326: delete "groups", the first one.

Response: Corrected as suggested
Comment: These are some examples of typos, but there are probably more. So, I strongly recommend doing a serious and thorough check

Response: I have thoroughly proofreaded whole manuscript for all kinds of typos errors and tried best to correct them